# School Feeding Programs: What Happens Globally?

**DOI:** 10.3390/ijerph19042265

**Published:** 2022-02-17

**Authors:** Alessandra Cupertino, Veronica Ginani, Ana Paula Cupertino, Raquel Braz Assunção Botelho

**Affiliations:** 1Department of Nutrition, College of Health Sciences, University of Brasilia, Brasilia 70910-900, Brazil; 2Master’s Programs Public Health, Department of Nutrition, College of Health Sciences, University of Brasilia, Brasilia 70910-900, Brazil; vcginani@gmail.com; 3Medical Center Wilmot Cancer, Wilmot Cancer Institute, University of Rochester, Rochester, NY 14627, USA; paula_cupertino@urmc.rochester.edu; 4Master’s Programs Nutrition, Department of Nutrition, College of Health Sciences, University of Brasilia, Brasilia 70910-900, Brazil; raquelbabotelho@gmail.com

**Keywords:** world food program, school feeding, children, countries, food security

## Abstract

School feeding programs (SFPs) are an important effort to address food insecurity, improve nutritional education, and ultimately improve health outcomes. The objective of this research was to describe the nutritional, cultural, food safety, and agro-family participation of SFPs in different countries and observe the SFP in low-middle and high-income countries to establish disparities. The study followed documentary research of SFP official online resources complemented by a literature review. The programs were assessed in four criteria: (1) nutritional aspects, (2) cultural aspects, (3) food safety, and (4) agro-family participation. Out of 192 countries registered, 117 countries (60.93%) have an SFP, and only 8 (4.16%) do not have SFPs. A total of 67 countries (34.89%) did not have an official online resource and did not respond to follow-up emails. Out of the 117 countries with SFP, all of them had a detailed description of their nutritional aspects, cultural aspects (11.96%), food safety (16.23%), and agro-family participation (23.93%). Europe and Central Asia were the continents with the most comprehensive SFP. While most countries incorporate nutritional aspects and healthy food, cultural, food safety, and agro-family participation must be incorporated in their SPF to optimize children and adolescent development. Moreover, the study identified disparities across countries where the SFPs were identified in low-middle countries compared to middle- and high-income countries.

## 1. Introduction

The biopsychosocial development of children (5 to 19 years old) is directly linked to adequate and healthy nutrition [1,2]. Unlike adults, healthy nutrition for children consists of understanding the differences existing in each stage of life, highlighting the first 1000 days. At his stage, nutritional requirements meet a rapid evolution of the nervous and immune system, and are essential in the formation of good eating habits. Thereafter, good nutrition will meet the child’s intellectual and physical development demands which accelerates again during adolescence [3].

Food and nutrition security for children is defined as the guaranteed right to access quality food, in a sufficient quantity, without compromising the access to order essential needs [4]. Despite the world economic development and established guidelines, more than 802 million people worldwide face food insecurity of which, 34.7 million were located in Latin America. Among children, the data show that at least one in three children do not have access to the food necessary for their health and physical and cognitive development [3]. Hence, it must be a priority for the government of different countries and a joint effort of all nations to guarantee its population’s food and nutritional security.

In addition, hunger is part of the modern world. More than 820 million people worldwide face food insecurity of which, 34.7 million are in Latin America. Eliminating hunger by 2030 is the United Nations goal linked to sustainable development (SDG) [5]. Food insecurity and suboptimal nutrition are also linked to obesity, an emerging trend in low-middle income countries [6]. About 40 million children under five years old are overweight in these countries, and about 120 million children and adolescents are obese [7].

Specifically, in Latin America and the Caribbean (LAC) countries, an epidemiological and nutritional transition is rapidly taking place with different changing patterns of obesity compared to developed countries [8]. In Brazil, the prevalence of obesity is around 20% in men and 20–30% in women. An urban population-based study in Argentina, Chile, and Uruguay (*n* = 7524) determined that the prevalence of obesity was 35.7%, and central obesity was 52.9% [9].

Combating the triple burden of malnutrition–undernutrition, hidden, hunger, and obesity that plagues children, especially in developing countries, depends partly on the balanced meals provided at school [10]. Children spend at least five hours at school, especially in low-middle income countries, and take their main meals [6]. According to the World Food Program (WFP) guidelines, a comprehensive school feeding program (SFP) is linked to a country’s financial allocation of public food and nutrition [5]. In Brazil, for example, public school meals are free, 100% subsidized by the government [11]. On the other hand, in the United States of America (USA), the meal is not charged depending on the family income to which the student belongs [6]. In Canada, in contrast, there is no national SFP. Only a few provincial programs are spread across the country [12].

Since 2006, the WHO has been providing a framework to ensure integrated school programs based on some initiatives in the school environment. The aim is to address the burden of nutrition-related health problems, overweight, obesity, and malnutrition. The initiative is focused on 26 criteria divided into five blocks: (1) school nutrition policy, (2) awareness and training of the school community, (3) curriculum base and health promotion, (4) favorable school environment for good nutrition, and (5) supportive school nutrition and health service [2]. Since then, this initiative has been used worldwide as a self-assessment tool for existing programs and as part of projects and research. 

A recent study in Brazil sought to identify the main criteria used in preparing school menus. They are habits, food culture, acceptance; nutritional characteristics; food availability; management and execution. Promoting a healthy environment, including food systems that promote a diverse, balanced and healthy diet, requires the involvement of multiple sectors and stakeholders, including the government and public and private sectors.

In a global setting, it is expected that almost every country provides food for their students on some scale, reaching around 368 million children [13]. Nevertheless, nutrition disparities across countries are complex underlying suboptimal school-based feeding programs that impact school performance and poor health outcomes [1,14,15]. These nutrition disparities within and across countries reinforce food insecurity and obesity classified as a global health priority. Consequently, it becomes essential that school feeding programs offer healthy and nutritionally adequate meals [16], and thus knowing what each program prioritizes and how it is organized is fundamental to aligning overall decisions on the issue in question.

This study is justified by the need to understand how school feeding programs are presented worldwide, having as a research question: What is the scenario of school feeding programs in the world to meet the premises of the World Health Organization for healthy eating? Therefore, the present documentary research aims to describe the presence of cultural and nutritional aspects, food safety, and the presence of agro-family in school-based feeding programs globally. We also aimed to observe the SFP in low-middle and high-income countries to establish disparities. Results will inform the nutrition disparities delivered in schools and guide future interventions designed to eliminate food insecurity and hunger globally.

## 2. Materials and Methods

The present study consists of a documentary analysis conducted through the consultation of the official websites followed by email and literature reviews. It refers to the analysis of documents containing information about the phenomenon studied [17]. Thus, it was elaborated to identify, verify, and assess documents relating to the School Feeding Program (SFP) in several countries to contextualize the facts and enable a global vision of the different realities experienced by peoples. The documents found were objectively analyzed in their original source, allowing the location, identification, organization, and evaluation of the information found. The research systematically took place concerning the capture of documents, processing, and analysis of their content [17,18]. Thus, we sought to reach a greater number of research sources to describe the presence or absence of SFP and information about them as the present characteristics. Data were collected to complement the SFP panorama in the world, identifying which countries have programs, and how they contribute to nutritional and food security. 

All 192 countries registered by World Health Organization [19] until September 2019 were included when searching for information. To conduct the study, all official websites in the countries were visited by the Google platform from September 2019 to January 2020. For each country, we sought to identify the presence or absence of an SFP based on the visit to the website. Website search utilized “School feeding program” keyword. When the country’s official website did not present any information, an email was sent with a cover letter for the research and a request for information about public relations, the country’s ministry of education, or a specific body. The letter asked the following question: Does this country have a school feeding program? Is it possible to receive information on it or a website?

When building the database, data were collected through the Google search to respond to the following question: “Does the country have a school feeding program”? For countries with an official language other than English, the search was performed in the country idiom relying on google translate. Two emails were sent to inquire about September 2019 over January 2020 to complete the database. Finally, the results from the website search and emails were classified dichotomically in “YES” or “NOT” for the presence of a school-based feeding program.

To complement the school-based feeding program database, a literature review was carried out in the following databases: PUBMED, LILACS, Scielo, Google academic, and Science Direct. The following descriptors were used to search for articles: “School feeding program”, “public policy”, “guideline” and “world” and their combination. The inclusion criteria for selecting articles were articles in English, Portuguese and Spanish, article in full that portrayed the theme of this research with open access, and articles published and indexed in these databases in the last five years.

To identify the structural characteristics of each program, we identified four aspects that add up to conceptualizing food security. In this way, we transcend the definition of healthy food, including, in addition to nutritional aspects, access, safety, and origin. For that, a search was also carried out on the websites of the agencies responsible for managing the program in each country to learn about the program and the official regulations. The programs were characterized according to the four criteria described by the World Health Organization in the concept of food security [7]: 

(1) presence of nutritional aspects: understanding that the nutritional aspects of food enable the child’s growth and development according to previously defined parameters; (2) presence of cultural aspects: valuing cultural aspects based on the presence of basic foods from the local food tradition and also considering the social and cultural aspects of the target population; (3) food safety: all procedures adopted throughout the food chain to ensure its safety; and (4) presence agro-family: comprising the participation of the agro-family in school menus, with food products supplied by local farmers. It was also noted, as a fifth item, the government´s participation, whether it is 100% or partially subsidized. The presence of any of these aspects in the SPF document justified their inclusion as a feature of the program.

Data on the presence or absence of the criteria were captured by information from official websites, email responses, or by searching for articles found about SFP. Figure 1 presents a flowchart of the steps developed.

The countries were grouped according to the World Food Program [12] which determined the distribution of the programs in the following regions: America/Caribbean; Africa/Middle East; Europe/Central Africa; and Southeast Asia/Pacific. In this way, the number of countries per region that presented each of the criteria in their respective program was registered [12]. Human development data from each country provided in the Human Development Report [20] were used to recognize existing disparities.

Descriptive statistics were used to analyze the results, determining the frequency of the characteristics of the programs related to their respective countries.

## 3. Results

In the present study, out of 192 countries registered in the WFP, 117 countries (60.93%) have a school-based feeding program (SPF), and only 8 (4.16%) did not have a program (Figure 2). A total of 67 countries (34.89%) did not have an official online resource, scientific manuscripts do not respond to follow-up emails.

The countries without a SFP were Angola, Cameroon, Greece, Iran, Iraq, Pakistan, Tuvalu, and Yemen. These countries are located explicitly in the following regions: Africa/Orient (*n* = 04); Southeast Asia/Pacific (*n* = 03); and Europe (*n* = 01). 

It is noteworthy here that only the countries that presented information or that information was found were quantified. Thus, the percentage was made according to the complete information. 

Out of the 117 countries with SPFs, all of them had a detailed description of their nutritional aspects (100%). However, 11.96% presented cultural aspects, 16.23% food safety, and 23.93% agro-family participation (Table 1). Europe and Central Asia had the most comprehensive school-based feeding programs.

Most countries did not report whether or not the government subsidizes school meals. However, among the countries that reported (*n* = 54), 87.0% subsidize 100% of food, and 13.0% partially subsidize. It is important to note that of the countries that subsidized 100%, 57.1% are classified by the United Nations as “high human development” countries.

## 4. Discussion

The results point to a worldwide trend of using SFPs as a strategy to guarantee food and nutrition security. However, there is still a large disparity among developed and developing countries. Even in the poorest countries where SFPs exist, the issues addressed are limited. It is observed that coverage is concentrated in countries with higher incomes, reinforcing the recent study “The Global Child Nutrition Foundation” [21]. The programs in these countries aim to meet nutritional goals, ensure food and nutritional security, promote an environment for students to work as a team, and strengthen social interaction. Specifically in Europe, the School Feeding Council discusses ways and means to provide healthy food in schools and all stakeholders’ involvement. For this, it promotes a discussion forum with representatives from all countries. The following aspects are discussed in these forums: nutrition and health of food at school, respect for eating habits, partnerships with private companies and NGOs (non-governmental organizations), and community involvement.

On the other hand, some countries in poverty do not benefit from the SFP. In low-income countries, the expectation is a large percentage of food insecurity in the population. According to the World Bank, 55 countries (28.64%) distributed in the world regions are considered low-middle income, and according to the present study, five of these low-income countries do not have SFPs (Figure 2) [22]. Most of them are regions of major political and economic conflicts, low-middle income countries where malnutrition is present and recognized as a challenge for public health [23]. It is known that the contexts of economic crisis, fluctuations in food prices, wars, political conflicts, and natural disasters deprive millions of people of accessing adequate food. Thus, there is a suboptimal investment in the health and nutrition of school-age children and adolescents, which is an obstacle to their development [5,14].

Another factor that enhances the situation of poverty and hunger, specifically in Africa, is the underreporting of malnutrition cases. School-age children are not commonly included in health and nutrition surveys. Therefore, an up-to-date overview of their nutritional status is not available [12,24]. According to WFP (2013) data [12], coverage of school feeding programs is still low in the regions with greater demand, similar to low-income countries [12,23]. Thus, the importance of a school feeding program in these regions may be catalyst for development [12]. On the other hand, middle-income countries represent 52% (*n* = 100) and high-income countries 19% (*n* = 37), being the countries that concentrate on school feeding programs. Therefore, school feeding programs are common in developed and developing countries with different scenarios. Furthermore, the aspects of food security analyzed in this research were present in a different way in the investigated countries. Next, each of the aspects and the possible implications of their presence or absence in SFPs will be addressed.

### 4.1. Nutritional Aspects

The nutritional aspect was covered by all the countries that reported having SFP. This characteristic of the programs highlights the relationship between food and health in a more direct way. Malnutrition, obesity, and chronic non-communicable diseases are important public health problems and should be highlighted. Nutritional aspects define the essential nutrients for a healthy diet and its nutritional balance. Therefore, the SFPs describe the nutritional characteristics of their meals, which should be based on WHO recommendations, respond to programs objectives, and align with national food guidelines if available [25]. Among the WHO recommendations, the following were mentioned in the analyzed SFP: insert vegetables in large meals, eat fresh fruit and raw vegetables in small meals, eat fruit and vegetables in varieties and in season; limit saturated fat intake to 10% of total daily and trans-fat intake to 1% of total daily energy; limit free sugar intake to less than 10% of total daily energy and further reduce free sugar to less than 5% of total daily energy for additional health benefits. Additionally, keep salt intake to less than 5 g/day [2]. Thus, the SFPs seek to ensure that all meals at school meet the children´s daily needs regarding the quality and quantity of energy and nutrients. Although, the SFPs identified in the present study focus on nutritional aspects they did not establish a professional nutrition program as a requirement. In Brazil, some studies have assessed the importance of the nutritionist in the management and implementation of their program “PNAE” [26,27], but few studies have evaluated the role of the nutritionist in planning school meals [28].

### 4.2. Cultural Aspects

In the present study, as seen in Table 1, only 37 countries (31.6%) present the cultural aspect as an important basis for school feeding in the characteristics of their programs. Considering the other characteristics analyzed in the programs, according to Botelho [28], healthy and adequate food must be based on eating practices that also consider the social and cultural aspects of the target population. Ginani [29,30] described the importance of regional foods and ingredients in the Brazilian territory. The author identified that menus need to be adequate according to regional and national guidelines that recommended the consumption of culinary preparations based on local foods. Cultural elements are associated with the acceptability of food and can preserve agricultural habits of sustainable systems, maintenance of biodiversity, and sustainability [30]. Enhancing the cultural aspects is important to encourage children to develop cooking skills in the school environment, thus presenting their culture and eating habits [2]. A study identified that the acceptability of school menus is related to regional products, that is, “local products” [31]. In this study, only one country presented a description of the school feeding program’s cultural characteristics and five countries presented agro-family characteristics. Especially in countries with large numbers of malnourished people and countries in poorer regions, locally produced food is crucial for economic development in the region. However, Table 1 shows that most countries that address the cultural aspect of menus in their programs are those with very high or high HDI. This fact can increase public health disparities, further distancing the most vulnerable populations from ideal situations.

### 4.3. Food Safety

Similar to the cultural aspects, only 41 countries (35.0%) address food safety in their SFPs. Worldwide, foodborne illnesses affect about 30% of the population every year [32]. In addition to the risk of death, foodborne illnesses can cause diarrhea, vomiting, abdominal pain, fever, and weakness in school-age children. These reactions can increase students’ absences from classes in the school environment affecting concentration and, consequently, school performance [33]. It is estimated that 600 million people worldwide get sick after eating contaminated food, 1 in 10 people. In addition, 420,000 die each year, resulting in the loss of 33 million years of healthy life. Epidemiological data reflect the importance of adopting strategies capable of mitigating this problem, especially in the production of meals for vulnerable communities, such as children, who are normally the most affected [33,34]. Few countries with medium or low HDI have contemplated these aspects. Outbreaks of foodborne illnesses can be fatal for populations with nutritional deficiencies or other comorbidities. For the child population, the neglect of this aspect is a generator of important social problems. Therefore, urgent strategies must be considered to modify this scenario.

### 4.4. Agro-Family

Finally, the inclusion of agro-family in the supply of schools is an important alternative to guarantee food security. Once again, less than half (41.8%; *n* = 49) of the schools that reported having SFPs mentioned privileging agro-family products (Table 1). Among countries, 79.6% are countries with very high or high HID. Food systems are processes that include agriculture, livestock, production processing, distribution, supply, marketing, preparation, and consumption of food and beverages [35]. The foods system needs to guarantee the integrity and maintenance of the planet’s biodiversity. That is, ensuring that the present natural resources are available in a way that enhance the right of future generations to also use them, meeting their needs and enabling a healthy quality of life. It is understood that an unsustainable environment is not capable of producing healthy food. The hegemonic development in the field based on agribusiness, on monoculture, and the use of pesticides has brought irreversible damage to the environment [36]. This described scenario has been causing negative impacts on health and the environment. Thus, this is the subject of one of the discussions in the world for the guarantee of the Sustainable Development Goals (SDGs) [37].

The environmental sustainability of food systems must be associated with the actions of individuals or groups about their food choices and the orientation of national food so that there are improvements in the population’s nutritional health [37]. The participation of family farming in the school feeding program creates market opportunities for small producers. In Brazil, there is a specific law [38], that requires that 30% of the amount transferred by the national school feeding program must be invested in the direct purchase of family farming products. This measure encourages economic development and the sustainable development of communities [39].

Thus, it contributes to ensuring food and nutritional security in the school environment and ensures better meals and healthier and more sustainable eating practices [2,7].

The quality of school feeding must be evaluated based on environmental and sustainability issues. Sustainability being one of the recommendations for school feeding in the world [40]. Thus, school feeding programs must support sustainable development by encouraging the purchase of local food produced by producers in the region [41]. Furthermore, good sustainability practices aim to solve part of the ecological impact caused by food production in food services [42].

In Ghana, the school feeding program has been redesigned since 2011. While the program’s goal was to boost local food production, farmers are not linked to the market created by the program. However, countries such as Ethiopia, Malawi, Mozambique, Nigeria, and Senegal have benefited from FAO and WFP implementation programs to link local agriculture with school feeding through specific food procurement tools [5,6,25,26,43].

### 4.5. Specific Characteristics of Some SFP

#### 4.5.1. Africa

In general, although it ranks second in the school feeding programs on the African continent, it is difficult to obtain information and details about them. Another relevant fact is the high incidence of malnutrition and food insecurity among school-age children. Therefore, the effectiveness of programs in achieving adequate and healthy school nutrition is questioned. Consequently, despite the existence of the programs, there are few references and documents that guide public policies. In other words, officially, some countries have programs, but paradoxically, “they do not exist”, since they are not instructed [12,44].

#### 4.5.2. Southeast Asia and Pacific

In Southeast Asia and the Pacific, 160 million children [6] in several countries receive school meals. All states and territories have implemented region-specific school food service policies in Australia. The first was developed in New South Wales (NSW) in 2005, while the most recent was implemented in Tasmania in 2014. The policies are mandatory for all government schools in every jurisdiction except Tasmania. A curiosity of the SFPs in this region is the use of educational tools to encourage healthy eating. Most regions use traffic light systems that categorize food and beverages into three groups: green, amber, and red. The green foods group is based on the five main food groups, fruits, vegetables, grains, meat, and dairy products, and they need to be consumed daily. The amber group has nutritional value with a moderate amount of energy and saturated fat, salt, and sugars. Finally, the red group has no favorable nutritional value and contributes to an excessive amount of energy, total fat, sodium, and sugars [6,45].

The eleven programs in this region only have nutritional characteristics in their description. The other characteristics were not found, even in Australia, which has implemented school feeding policies. The other features need to be incorporated into programs to better link local food production and the school.

### 4.6. School Feeding Programs’ Challenges

It is understood that the world faces somewhat opposite problems: (1) an increase in obesity in developed countries [46] and (2) food shortages and malnutrition in developing countries. Hunger is an ongoing problem that affects more than 1.2 billion people who do not have enough to eat in the world today [47]. Reducing obesity and malnutrition are complex challenges that can be better managed by school feeding programs [6,47].

In this sense, worldwide attempts to direct the consumption of healthy foods occur regularly. An example is that WHO has been encouraging countries to implement nutrition policies that specify what foods and beverages can and cannot be offered in schools. Although some programs have been in existence for more than fifty years, such as the PNAE in Brazil and the NSLP in the USA, it was only from the 2000s that school feeding attracted the interest of government agencies at international, national, and regional levels. The government interest is justified by considering school feeding as a way to combat nutritional inadequacy in adulthood, ecological threats (excessive consumption waste), social challenges (food crisis, scarcity), and cultural issues (debates and conflicts around beliefs and values and cultural issues) [48].

## 5. Conclusions

With this study, an overview of the school feeding programs in the world was identified. The region with the greatest coverage of programs was Europe/Central Asia. Although these programs contemplate the nutritional aspects for healthy school feeding, other criteria such as culture, food safety, and agro-family need to be revised to comply with the FAO and WHO recommendations. The America/Caribbean regions stand out here. Despite encompassing the largest and oldest program globally, the NSLP in the US and PNAE in Brazil need to be further explored to meet the objectives of adequate school feeding.

The results obtained from the study did not take into account aspects such as the size of the country, population, and the number of countries in the analyzed regions. It is concluded that there are still many countries without information, either because of the difficulty in finding them on the official site, without updating at the time of the search or because of the lack of scope of the bibliographic search.

## Figures and Tables

**Figure 1 ijerph-19-02265-f001:**
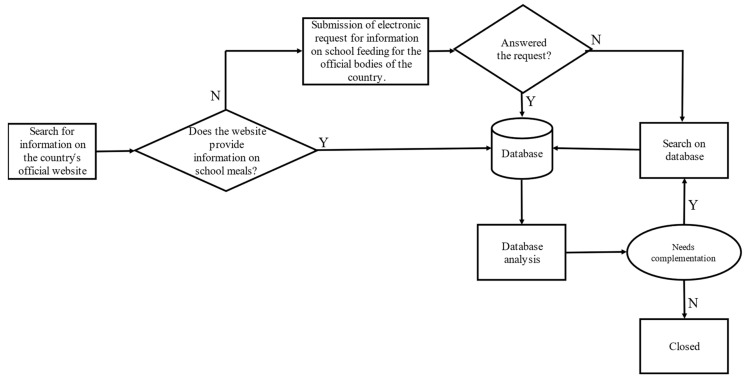
The flowchart of the steps developed for the documentary research.

**Figure 2 ijerph-19-02265-f002:**
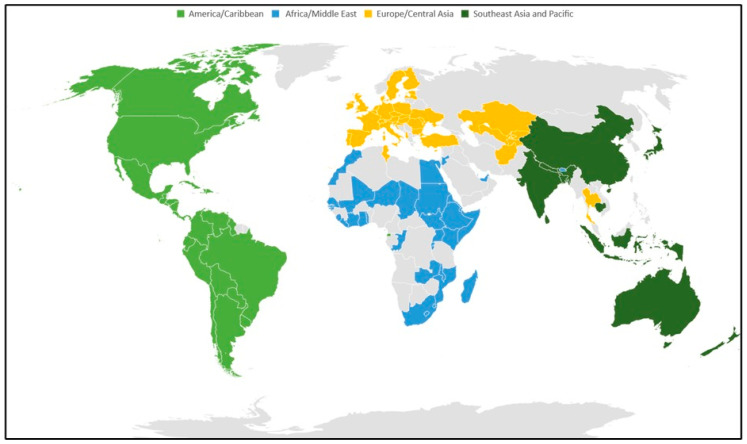
Distribution of school-based feeding programs by regions and their respective countries. America/Caribbean (*n* = 28; 23.91%): Argentina, Bahamas, Barbados, Bolívia, Brazil, Canada, Chile, Colombia, Costa Rica, Cuba, Dominica, Dominican Republic, Ecuador, El Salvador, Equatorial Guinea, Guatemala, Guyana, Haiti, Honduras, Mexico, Nicaragua, Panama, Paraguay, Peru, Trinidad and Tobago, USA, Uruguay, and Venezuela. Africa/Middle East (*n* = 38; 32.5%): Bhutan, Burkina, Burundi, Faso, Cape Verde, Chad, Congo, Côte d’Ívoire, Djibouti, Egypt, Eritrea, Essuatíni, Ethiopia, Gambia, Ghana, Guinea Bissau, Jordan, Kenya, Lesotho, Liberia, Madagascar, Malawi, Mali, Morocco, Mozambique, Republic of Niger, Rwanda, São Tomé and Príncipe, Senegal, Sierra Leone, Somalia, South Africa, South Sudan, Sudan, Togo, Uganda, United Arab Emirates, and Zambia. Europe/Central Asia (*n* = 41; 35%): Afghanistan, Albania, Andorra, Armenia, Austria, Azerbaijan, Belgium, Bulgaria, Croatia, Cyprus, Czech Republic, Denmark, Estonia, Finland, France, Germany, Hungary, Ireland, Italy, Kazakhstan, Kyrgyzstan, Latvia, Luxembourg, Netherlands, Macedonia, Poland, Portugal, Moldova, Romania, Slovakia, Slovenia, Spain, Sweden, Switzerland, Tajikistan, Thailand, Tunisia, Turkey, Ukraine, United Kingdom, and Uzbekistan. Southeast Asia/Pacific (*n* = 11; 9.4%): Australia, Bangladesh, Cambodia, China, India, Indonesia, Japan, Nepal, New Zealand, Sri Lanka, and East Timor.

**Table 1 ijerph-19-02265-t001:** Information provided by countries on the presence of a school feeding program (SFP), local government participation in its funding (partial or total subsidy for school feeding), and the presence of nutritional, cultural, food safety, and agro-family aspects in accordance with local human development.

Human Development Index (HDI) (UNDP, 2020)	Countries that Reported Having SFP (%)	Countries that Did Not Report Having or Do Not Have SFP (%)	Countries that Reported Having a Government Subsidy for School Meals (%)		Aspects of Food Security Addressed in the SFP (%)			
			Partially	Totally	Nutrition	Culture	Food Safety	Agro-Family
*n*	117	75	47	7	117	37	41	49
Very high human development	34.2	32.0	57.4	14.3	34.2	62.2	56.1	49.0
High human development	26.5	26.7	27.7	57.1	26.5	29.7	31.7	30.6
Medium human development	18.8	20.0	8.5	28.6	18.8	5.4	7.3	12.2
Low human development	19.7	12.0	6.4	0.0	19.7	2.7	4.9	8.2

## Data Availability

World Health Organization (WHO). Alphabetical List of WHO Member States; WHO: Geneva, Switzerland, 2019.

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
