# Peer review of "School Feeding Programs: What Happens Globally?"

_ijerph, 2022, doi:10.3390/ijerph19042265_

Round 1
Reviewer 1 Report
The section results and conclusion is not adequate. The research question has to much more clear, further the discussion lacks of some cross-references and it is not clear how results and disussion are linked.
The basic assumption must also be questioned. Why are the school food programmes selected? This would have to be looked at more comprehensively.
Needs major revision
Author Response
Dear Editor,
Please see below the responses to reviewers' comments. In the revised manuscript, all the changes are highlighted using the "Track Changes" function in Microsoft Word.
Thank you for the opportunity to improve our manuscript!
Reviewer #1:
The section results and conclusion is not adequate. further the discussion lacks of some cross-references and it is not clear how results and discussion are linked
R: We made some changes in the discussion section to improve the link between references and relate to results.
The research question has to much more clear.
R: We included a research question in the manuscript.
The basic assumption must also be questioned. Why are the school food programmes selected? This would have to be looked at more comprehensively.
R: We included a sentence reinforcing the importance of school feeding programs and therefore the choice of the Search topic.
Reviewer #2:
In my opinion, after the review of the state of knowledge in the Introduction, it would be worth formulating a research problem. The research problem should result from a knowledge review indicating an insufficient state of knowledge in the area in question. In the Introduction chapter, the authors presented general issues that need to be solved, but it would be worth calling them a research problem. Only the formulation of the research problem should be a premise to identify a gap in the current state of knowledge and to present the purpose of the research
R: We included a research question in the manuscript.
In my opinion, the purpose of the article presented in the Abstract and in the Introduction is slightly different. In the Abstract, the Authors indicated that "The objective of this review was to describe nutritional, cultural ...", while in the Introduction one can read: "The aim of the present documentary research is to assess the SFP programs ...". I think it would be worth considering the same wording of the research goal set both in the Abstract and in the last paragraph of the Introduction chapter.
R: We apologize for the mistake, and we correct the objective.
Generally, in the case of the formulated research goal, it would be worth pointing to a cognitive (scientific) goal and a utilitarian (useful) goal at the same time. Even if the authors specified the purpose(s) of the research, it is worth defining unequivocally which of them are cognitive and which are utilitarian.
R: This study is justified by the need to understand how school feeding programs are presented worldwide to meet the premises of the World Health Organization for healthy eating. We included the importance of the study at the end of the introduction section.
The caption to the drawing does not say "Figure 1 presents a flowchart of steps developed" (line: 140). I suggest that you follow the wording provided in Supplementary Materials.
R: We apologize for our mistake and corrected it.
If the Authors analyzed nutritional aspects, cultural aspects, food security, and agro-family participation, it would be worthwhile to write in the Materials and Methods chapter what exactly is included in these concepts / scopes of analysis. Or, it is worth writing down what information the Authors were looking for in the detailed areas of the analysis. Although the descriptions of these detailed issues were developed in the Discussion (chapter 4), it is worth presenting the authors' assumptions in the Materials and Methods chapter, which could be included in nutritional aspects, cultural aspects, food security, and agro-family participation.
R: We included more information on the methods section for better comprehension.
I would like to know what are the requirements / standards / recommendations for daily nutritional needs for school children. I think that it would be worthwhile to provide the daily nutritional requirements of young people, expressed in MJ / person, taking into account the age ranges. This information can be provided in the Materials and Methods chapter. Based on the caloric content of daily meals (food rations), including those included in programs (e.g. SFP) and information on daily food requirements (in MJ / day / person), it would be possible to more reliably assess the value of nutritional programs.
R: The objective of our review was to find the school feeding programs and search for structural characteristics of the existing programs. We believe that another study could compare the nutritional requirements and discuss all the differences among the programs that present these standards. Programs are based on specific populations and need to be carefully analyzed in a research article.
I do not understand what the Authors wanted to convey in the paragraphs Figure S 2 (line: 200), Figure S 2 (line: 207) and Figure S 3 (line: 210). Are these any analysis results, suggestions or suggestions for summarizing the analysis results? Or is this a brief description of the various scenarios mentioned at the end of the paragraph on lines 198-199? If these are descriptions for scenarios, it might be worthwhile to name them differently, instead of Figure. I associate the word Figure with a drawing and I was looking for such drawings in the article, but I did not find them. It is true that Supplementary Materials wrote about the terms Figure S1, Figure S2 and Figure S3, but in my opinion it is necessary to clearly write in lines 200 and 210 that Figure S2 and Figure S3 can be found in additional materials. By the way, it is spelled "Supplementary" and not "Suplementary" as the authors wrote in line 369.
R: We apologize for our mistake. The titles of the Supplementary figures were inside the manuscript by mistake. We already withdrew. We decided to include the figure and Table 1 inside the manuscript and not as a supplementary file.
I do not understand why on line 219 the authors gave citation [2][2]. Is it citing the same source twice in the same place?
R: It was a mistake. We corrected it.
On line 245 the authors wrote "... As seen in table 3 ...", but I couldn't find table 3 anywhere. Please explain the information on table 3.
R: We apologize for our mistake and corrected it.
I don't know what is "Lei nº 11.947, de 16/6/2009" (line: 291). Please explain. I did not find this information (citations?) in References, hence my request for clarification.
R: We apologize and inserted the reference relating to Brazilian law.
In the References section, if articles in Portuguese and other languages are included, for example under numbers 27 and 28, then in my opinion you should also include the titles of the articles in English, and finally, write (in Portuguese). The same remark applies to other articles in languages other than English.
R: We translated the references as suggested.

Reviewer 2 Report
In my opinion, after the review of the state of knowledge in the Introduction, it would be worth formulating a research problem. The research problem should result from a knowledge review indicating an insufficient state of knowledge in the area in question. In the Introduction chapter, the authors presented general issues that need to be solved, but it would be worth calling them a research problem. Only the formulation of the research problem should be a premise to identify a gap in the current state of knowledge and to present the purpose of the research. In my opinion, the purpose of the article presented in the Abstract and in the Introduction is slightly different. In the Abstract, the Authors indicated that "The objective of this review was to describe nutritional, cultural ...", while in the Introduction one can read: "The aim of the present documentary research is to assess the SFP programs ...". I think it would be worth considering the same wording of the research goal set both in the Abstract and in the last paragraph of the Introduction chapter.
Generally, in the case of the formulated research goal, it would be worth pointing to a cognitive (scientific) goal and a utilitarian (useful) goal at the same time. Even if the authors specified the purpose(s) of the research, it is worth defining unequivocally which of them are cognitive and which are utilitarian.
The caption to the drawing does not say "Figure 1 presents a flowchart of steps developed" (line: 140). I suggest that you follow the wording provided in Supplementary Materials.
If the Authors analyzed nutritional aspects, cultural aspects, food security, and agro-family participation, it would be worthwhile to write in the Materials and Methods chapter what exactly is included in these concepts / scopes of analysis. Or, it is worth writing down what information the Authors were looking for in the detailed areas of the analysis. Although the descriptions of these detailed issues were developed in the Discussion (chapter 4), it is worth presenting the authors' assumptions in the Materials and Methods chapter, which could be included in nutritional aspects, cultural aspects, food security, and agro-family participation.
I would like to know what are the requirements / standards / recommendations for daily nutritional needs for school children. I think that it would be worthwhile to provide the daily nutritional requirements of young people, expressed in MJ / person, taking into account the age ranges. This information can be provided in the Materials and Methods chapter. Based on the caloric content of daily meals (food rations), including those included in programs (e.g. SFP) and information on daily food requirements (in MJ / day / person), it would be possible to more reliably assess the value of nutritional programs.
I do not understand what the Authors wanted to convey in the paragraphs Figure S 2 (line: 200), Figure S 2 (line: 207) and Figure S 3 (line: 210). Are these any analysis results, suggestions or suggestions for summarizing the analysis results? Or is this a brief description of the various scenarios mentioned at the end of the paragraph on lines 198-199? If these are descriptions for scenarios, it might be worthwhile to name them differently, instead of Figure. I associate the word Figure with a drawing and I was looking for such drawings in the article, but I did not find them. It is true that Supplementary Materials wrote about the terms Figure S1, Figure S2 and Figure S3, but in my opinion it is necessary to clearly write in lines 200 and 210 that Figure S2 and Figure S3 can be found in additional materials. By the way, it is spelled "Supplementary" and not "Suplementary" as the authors wrote in line 369.
I do not understand why on line 219 the authors gave citation [2][2]. Is it citing the same source twice in the same place?
On line 245 the authors wrote "... As seen in table 3 ...", but I couldn't find table 3 anywhere. Please explain the information on table 3.
I don't know what is "Lei nº 11.947, de 16/6/2009" (line: 291). Please explain. I did not find this information (citations?) in References, hence my request for clarification.
In the References section, if articles in Portuguese and other languages are included, for example under numbers 27 and 28, then in my opinion you should also include the titles of the articles in English, and finally write (in Portuguese). The same remark applies to other articles in languages other than English.
Author Response

(The authors gave the same response as above.)

Round 2
Reviewer 2 Report
Thank you for the changes and corrections made to the article, taking into account the suggestions presented in the review.